# Physiological and Transcriptomic Responses of *Illicium difengpi* to Drought Stress

**Baoyu Liu** [1,†], **Huiling Liang** [1,†], **Chao Wu** [1], **Xiyang Huang** [1], **Xiangying Wen** [2], **Manlian Wang** [1,*] and **Hui Tang** [1,*]

1   Guangxi Key Laboratory of Plant Functional Phytochemicals and Sustainable Utilization, Guangxi Institute of Botany, Guangxi Zhuang Autonomous Region and Chinese Academy of Sciences, Guilin 541006, China; lby@gxib.cn (B.L.); lhl@gxib.cn (H.L.); wuchao@gxib.cn (C.W.); hxy@gxib.cn (X.H.)
2   South China Botanical Garden, Chinese Academy of Sciences, Botanic Gardens Conservation International (BGCI) China Office, 723# Xingke Road, Tianhe District, Guangzhou 510650, China; wx-ying@scbg.ac.cn
*   Correspondence: wangml1978@163.com (M.W.); th@gxib.cn (H.T.)
†   These authors contributed equally to this work.

**Abstract:** *Illicium difengpi* Kib and Kim, an endangered plant unique to karst areas in China, has evolved an extremely high tolerance to arid environments. To elucidate the molecular mechanisms of the response to drought stress in *I. difengpi*, physiological index determination and transcriptome sequencing experiments were conducted in biennial seedlings grown under different soil moisture conditions (70~80%, 40~50% and 10~20%). With increasing drought stress, the leaf chlorophyll content decreased, while the proline (Pro), soluble sugar (SS) and malondialdehyde (MDA) contents increased; superoxide dismutase (SOD) and peroxidase (POD) activities also increased. Transcriptome sequencing and pairwise comparisons of the treatments revealed 2489, 4451 and 753 differentially expressed genes (DEGs) in CK70~80 vs. XP40~50, CK70~80 vs. XP10~20 and XP40~50 vs. XP10~20, respectively. These DEGs were divided into seven clusters according to their expression trends, and the Gene Ontology (GO) and Kyoto Encyclopedia of Genes and Genomes (KEGG) pathway enrichment results of different clusters indicated that genes in the hormone signal transduction and osmotic regulation pathways were greatly activated under mild drought stress. When drought stress increased, the DEGs related to membrane system and protein modification and folding were all upregulated; simultaneously, chitin catabolism- and glycolysis/gluconeogenesis-related genes were continuously upregulated throughout drought stress, while the genes involved in photosynthesis were downregulated. Here, 244 transcription factors derived from 10 families were also identified. These results lay a foundation for further research on the adaptation of *I. difengpi* to arid environments in karst areas and the establishment of a core regulatory relationship in its drought resistance mechanism.

**Keywords:** *Illicium difengpi*; drought stress; transcriptome sequencing; physiological characteristics; transcription factors; differentially expressed genes



## 1. Introduction

Drought stress, which can occur in all environments, regardless of borders and with no clear warning, can be caused by various factors such as global warming, rainfall anomalies and shifts in monsoon patterns, and has serious effects on plant physiology, biochemistry and gene expression [1]. Generally, drought stress significantly decreases the chlorophyll content, photosynthesis and the membrane stability index of plants while increasing the reactive oxygen species (ROS) content, osmolyte content, malondialdehyde (MDA) content and antioxidant enzyme activity [2,3]. In addition, the expression of genes responsible for drought tolerance in plants is induced by drought stress, especially those related to hormone (abscisic acid, cytokinin, ethylene, etc.) synthesis and signal transduction, osmolyte synthesis, antioxidant detoxification and ion transport [4].

*Illicium difengpi* Kib and Kim, of the family Octagonaceae, grows as a shrub or small tree with a height of 1~3 m, mainly on the top of karst limestone mountains in Guangxi, China [5]. Its distribution region ranges from 22°18'15.4″ N~25°2'33″ N and 106°1'39.6″ E~108°46'20.6″ E, and its vertical distribution covers altitudes of 450~1200 m [6]. Guangxi is a province in southwestern China located in the tropical and subtropical monsoon climate zone, which is characterised by distinct dry and rainy seasons. As shown in a previous investigation [6], the areas where *I. difengpi* is distributed have annual rainfall ranging from 1338.4 to 1606.8 mm. The rainy season is mainly concentrated from May to August, accounting for more than 65% of the annual rainfall, while there is little rainfall from late autumn to early spring. The soil in karst areas is shallow and dominated by rendzina, with high nitrogen and calcium levels, rich organic matter, and extremely low phosphorus and potassium levels. Uneven rainfall interacts with shallow soil and severe karst leakage in karst areas, leading to *I. difengpi* facing seasonal droughts year-round. Considering the intermittent drought conditions faced by the species, we carried out research on the effect of soil moisture on *I. difengpi* seed germination [7]. The results showed that the germination rate of *I. difengpi* seeds was the highest (≥70%) under 70% soil moisture (control). When the soil moisture content was below 40%, the seeds failed to germinate, but they resumed germination after rewatering (although the germination rate was less than 40%), and the start time and duration of germination were significantly shortened compared with the control. This result indicated that *I. difengpi* seeds have strong drought resistance and could germinate rapidly in the rainy season; thus, the species is a candidate tree for ecological restoration in karst areas.

However, *I. difengpi* is a traditional Chinese medicine used for treating rheumatic arthralgia, lumbar muscle strain and other diseases and is often used by humans [5]. In recent years, due to the excessive destruction of its native habitat by humans, coupled with the low natural reproduction rate of *I. difengpi* caused by the harsh environment, the wild germplasm resources of this species have continuously declined, and it is on the verge of extinction in many areas. Currently, wild *I. difengpi* plants can only be seen on the tops of a few karst rocky mountains and rarely grow on the mountainside [6]. The populations are stellately distributed, and the number of individuals in a single population ranges from a few to a dozen. This will make it difficult to exchange genetic material between wild populations, which may lead to a further decline in reproductive rates.

Previous studies on *I. difengpi* have mostly addressed the extraction and pharmacological activity of the chemical components of its bark. Dozens of chemical components, such as phenylpropanoids [8], aromatic glycosides [9], triterpenes [10], sesquiterpenes [11] and volatile oils [12], have been isolated from *I. difengpi* thus far. However, *I. difengpi* is an endangered species. Researchers should look for ways to protect the species while tapping into its medicinal value. Therefore, it is particularly important to study the physiological changes and molecular regulation mechanisms that appear in the species under stress conditions. Several studies on in vitro preservation and the effects of drought stress, soil environment and light intensity on the ecophysiological characteristics of *I. difengpi* have been carried out [13–17]. These studies laid a theoretical foundation for the artificial cultivation and germplasm resource protection of *I. difengpi*, but the molecular regulation mechanisms adopted by the species in response to drought stress have not yet been reported. In this study, transcriptome sequencing technology was used to analyse the expression patterns of drought resistance genes in *I. difengpi* under drought stress to lay a foundation for further revealing the molecular mechanism of adaptation to drought in the species.

## 2. Materials and Methods

### 2.1. Materials and Drought Treatment

The tested biennial seedlings were germinated from seeds collected from the same wild population in Jingxi County, Baise City, Guangxi. Before the experiment, monochronogenous soil collected around *I. difengpi* plants was prepared, and the soil moisture content was measured by the oven drying method (105 °C, 8 h). Then, the soil was put into nursery

bags, 2.0 kg per bag, for a total of 90 bags. The plants were transplanted into nursery bags filled with the soil and then placed in a greenhouse for recovery growth for 4 weeks, during which time 70~80% of the saturated water content of the soil was maintained (the saturated water content of the soil was 57%). During the experiment, the plants were randomly divided into 3 groups with 30 plants in each group, and the water supply was then stopped to naturally desiccate the plants. When the soil moisture contents of the 3 groups reached 70~80% (control), 40~50% (mild stress), and 10~20% (severe stress) of the saturated water content, they were measured by the weighing method every day to replace water losses to maintain the indicated level of stress. Leaves of the plants were used to conduct physiological indicators determination and transcriptome sequencing (Figure 1).

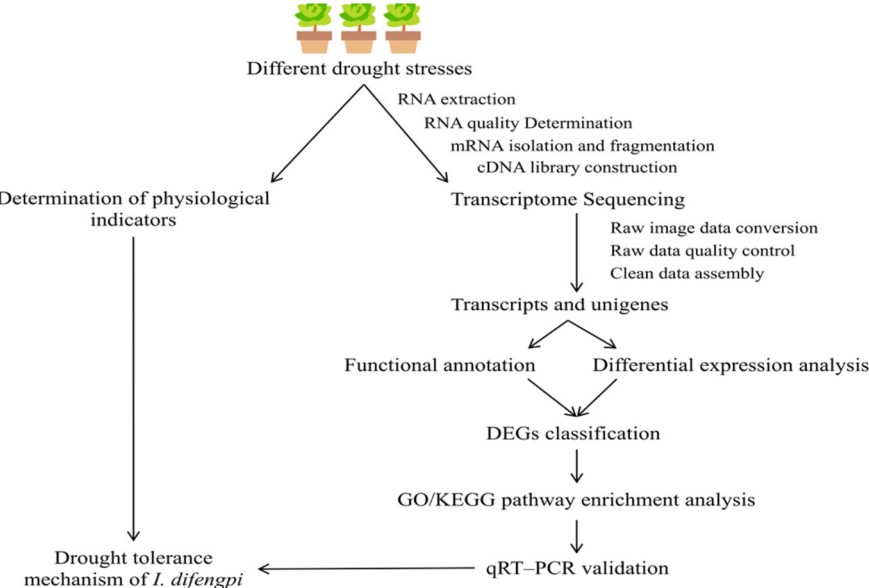

**Figure 1.** Flowchart of physiological indicators determination and transcriptome sequencing. DEG, differentially expressed genes; GO, gene ontology; KEGG, Kyoto Encyclopedia of Genes and Genomes; qRT-PCR, quantitative real-time PCR.

### 2.2. Determination of Physiological Indicators

The determination of various physiological indicators was carried out according to the methods of Li [18], and fully expanded leaves were the object of the experiments, with three biological repetitions for each treatment. The main instruments used in the experiment were a UV spectrophotometer (TU-1901, Persee, Beijing, China), high-speed centrifuge and water bath.

The photosynthetic pigment content was measured using the ethanol extraction method. Cleaned leaves (0.25 g each plant) detached from different plants were placed in 25 mL volumetric flasks, and then 95% ethanol was added until the liquid level reached the scale mark. The volumetric flasks were placed under darkness for 48 h, during which period they were shaken 2–3 times. Then, the absorbance of the extracts was measured at 665, 649 and 470 nm. The levels of chlorophyll a (Chl a), chlorophyll b (Chl b) and carotenoids (Car) were calculated according to the following formulas:

$$Chl\ a = 13.95 \times A_{665} - 6.88 \times A_{649}$$

$$Chl\ b = 24.96 \times A_{649} - 7.328 \times A_{665}$$

$$Car = (1000 \times A_{470} - 2.05 \times Chl\ a - 114.8 \times Chl\ b)/245$$

where $A_{665}$ is the value of the absorbance of the extracts at 665 nm, $A_{649}$ represents the absorbance of the extracts at 649 nm, and $A_{470}$ stands for the absorbance of the extracts at 470 nm.

The proline (Pro) content was determined via the acidic-ninhydrin method. A total of 0.3 g of cleaned fresh leaves of each plant was used as the test sample, which was then cut into small pieces and extracted with 5 mL of 3% sulfosalicylic acid in a boiling water bath for 10 min. After filtering, 2 mL of each extract was held for 30 min in boiling water after treatment with 2 mL ninhydrin and 2 mL glacial acetic acid, which was followed by adding 4 mL toluene. Subsequently, the mixture was shaken, allowed to stand and centrifuged (3000 r/min, 5 min). The proline content of the extract was measured at 520 nm and calculated against the standard proline curve (the standard curve concentration gradients were 0, 1.0, 2.0, 3.0, 4.0, 5.0 mg·mL$^{-1}$). Last, the proline content of the fresh sample was calculated according to the formula proposed by Li [18].

The soluble sugar content was estimated by the anthrone colourimetric method. Leaves collected from different plants were cut into pieces after cleaning, and 0.3 g of the pieces per serving was used for soluble sugar extraction by adding 5 mL distilled water and boiling in a water bath for 30 min (cycle twice). Water extracts were collected in 25 mL volumetric flasks and brought up to a fixed volume with distilled water to obtain the test solutions. The test solutions (0.5 mL) were used for the colourimetric reaction by mixing with distilled water (1.5 mL), anthrone ethyl acetate (0.5 mL, 1 g anthrone dissolved in 50 mL ethyl acetate) and sulfuric acid (5 mL, with a specific gravity of 1.84). The mixtures were placed in a boiling water bath for 1 min. The light absorption of the mixtures was estimated at 630 nm. The levels of soluble sugar were calculated using a sucrose standard following the formula proposed by Li [18].

The nitro blue tetrazolium (NBT) reduction method was used to determine the activity of superoxide dismutase (SOD). The enzyme extracts were prepared by grinding the cleaned fresh leaves (0.5 g) with phosphate buffer (1 mL, 0.05 mol·L$^{-1}$, pH 7.8), after which homogenates were diluted (4 mL, phosphate buffer) and centrifuged (1000 r min$^{-1}$, 20 min, 4 °C) to obtain the supernatants. SOD activity was determined for the assay tubes, control tubes and blank tubes. The reaction mixture in the assay tubes consisted of phosphate buffer (1.5 mL), methionine (0.3 mL, 130 mmol·L$^{-1}$), NBT (0.3 mL, 750 µmol·L$^{-1}$), EDTA-Na$_2$ (0.3 mL, 100 µmol·L$^{-1}$), riboflavin (0.3 mL, 20 µmol·L$^{-1}$), enzyme extracts (0.05 mL), and distilled water (0.25 mL). The components in the control tubes and blank tubes were the same as those in the assay tubes except that the enzyme was replaced with phosphate buffer. Then, the assay tubes and control tubes were illuminated under light of 4000 flux for 20 min, while the blank tubes were subjected to dark treatment. The light absorption of the assay tubes and control tubes was measured at 560 nm, and one unit of SOD activity was considered the amount of enzyme used for 50% inhibition of the NBT reduction reaction.

Peroxidase (POD) activity was determined by the guaiacol method. Leaves (0.5 g per sample) detached from different plants were ground into homogenates by adding phosphate buffer (2 mL, 0.05 mol·L$^{-1}$, pH 5.5), and then the homogenates were centrifuged (5000 r·min$^{-1}$, 10 min, 4 °C) and the supernatants were collected, while the precipitates were extracted twice again with phosphoric buffer (4 mL). Subsequently, all extracts were collected in 25 mL volumetric flasks and brought to volume, and finally, the test enzyme extracts were obtained. The POD activity assay reaction system consisted of phosphate buffer (2.9 mL), H$_2$O$_2$ (1 mL, 2%), guaiacol (1 mL, 0.05 mol·L$^{-1}$) and enzyme extract (0.1 mL). The mixture was held for 15 min in a 37 °C water bath, and then trichloroacetic acid (2.0 mL, 20%) was added to terminate the reaction. The blank was run in the same manner, but the enzyme extract was inactivated by boiling for 5 min before adding to the reaction system. The absorbance value of the product at 470 nm was measured and was used to calculate the POD activity following the formula proposed by Li [18].

The malondialdehyde (MDA) content was estimated with the thiobarbituric acid test (TBA). The tested sample was prepared by grinding the leaf sample (0.5 g) in trichloroacetic acid (5 mL, 5%), followed by centrifugation (5000 r·min$^{-1}$, 10 min). The supernatant was collected, and thiobarbituric acid was added. Then, the mixture was placed in a boiling water bath for 30 min. The reaction product was used to determine the absorbance values

at 450, 532, and 600 nm, and the values were used to calculate the MDA content following the formula proposed by Li [18].

### 2.3. Transcriptome Sequencing

#### 2.3.1. RNA Extraction, Library Construction and Sequencing

Three biological duplicates were randomly selected from each treatment, and their leaves were collected. The obtained leaves were quickly frozen with liquid nitrogen and then stored at $-80°C$ for RNA-seq analysis. The samples were sent to Major Biomedical Technology Co., Ltd. (Shanghai, China) for total RNA extraction and transcriptome sequencing after packaging in a dry ice box. RNA extraction was performed with TRIzol® Reagent (Invitrogen, San Diego, CA, USA), following the manufacturer's instructions; RNA purity and concentration were checked using a Nanodrop 2000 with the thresholds of $C \geq 200$ ng/μL and $1.8 \leq OD\ 260/280 \leq 2.2$; and RNA integrity number (RIN) values were assessed by using an Agilent 2100 system. Oligo (dT) magnetic beads were used to isolate the mRNA from the total RNA, and the mRNA was then fragmented into short segments of approximately 200 bp by using a fragmentation buffer. Double-stranded cDNA was synthesised by reverse transcription using the mRNA segments as templates. Subsequently, end repair and single nucleotide A (adenine) addition were conducted, and cDNA libraries were then constructed via PCR enrichment. Finally, the cDNA libraries were detected with TBS380 Picogreen, and the qualified libraries were used for high-throughput sequencing (Illumina HiSeq 4000).

#### 2.3.2. Raw Data Processing

In order to facilitate the analysis, publication and sharing of sequencing data, the raw image data obtained from Illumina sequencing were converted into FASTQ format sequence data (raw data) through base calling [19]. To ensure the accuracy of the subsequent bioinformatics analysis, SeqPrep (https://github.com/jstjohn/SeqPrep, accessed on January 2019) was used to remove adaptor sequences, low-quality reads, and sequences with a high N rate and those shorter than 20 bp from the raw data; finally, clean data were obtained. Then, the base error rate, Q20, Q30 and GC content of the clean data were calculated. The clean data were assembled into nonredundant transcripts and unigenes using Trinity software (http://trinityrnaseq.sourceforge.net/ accessed on January 2019) [20].

#### 2.3.3. Annotation and Gene Expression Analysis

To explore the biological functions of the assembled transcripts and unigenes, they were annotated using BlastX searches and the following databases: NCBI nonredundant protein sequences (Nr), STRING, SwissProt, Pfam and Kyoto Encyclopedia of Genes and Genomes (KEGG) (E-value $< 1 \times 10^{-5}$) [21,22]. The sequencing reads were mapped to the reference transcriptome by using RSEM. The levels of gene expression were evaluated according to FPKM values (Fragments per Kilobase of transcript per Million mapped reads), and the specific calculation formula was as follows:

$$FPKM = 10^9 C/(NL) \tag{1}$$

where C represents the number of fragments that are uniquely aligned to the gene, N is the total number of fragments that are uniquely aligned to the reference gene, and L is the length of the gene [23].

#### 2.3.4. Screening and Enrichment Analysis of DEGs

Differentially expressed gene (DEG) screening was carried out with the software edge R (http://www.bioconductor.org/packages/2.12/bioc/html/edgeR.html, accessed on January 2019), a Bioconductor software package for examining the differential expression of replicated count data with the method based on the negative binomial distribution [24,25], according to the screening criteria of a false discovery rate (FDR) $< 0.05$ and a $|\log_2 FC| \geq 1$. In order to study the differences in molecular mechanisms of *I. difengpi*'s response to

different drought stresses, the DEGs were divided into different clusters according to their expression trends during the drought stress experiments. Subsequently, the GO (Gene Ontology, http://www.geneontology.org/ accessed on January 2019) and KEGG databases were used to perform the enrichment analysis of the DEGs in different clusters.

2.3.5. Quantitative Real-Time PCR (qRT–PCR) Analysis

In order to verify the accuracy of the RNA-seq data, a total of 6 DEGs that may be related to drought stress were selected to perform qRT–PCR analysis, which was conducted by Major Biomedical Technology Co., Ltd. (Suzhou, China). The experiments were run on the fluorescence quantitative PCR instrument (ABI7500, Applied Biosystems, Foster City, CA, USA) using ChamQ SYBR Color qPCR Master Mix (2×) (Vazyme Biotech Co., Ltd., Nanjing, China) according to the manufacturer's instructions. Each 20 μL fluorescent quantitative reaction system contained 16.4 μL of ChamQ SYBR Color qPCR Master Mix (2×), 1.6 μL of primers (5 μM) and 2 μL of cDNA template. The amplification program was set as follows: 95 °C for 5 min followed by 40 cycles of 95 °C for 5 s and 55 °C for 30 s, with a final step of 72 °C for 40 s. Relative expression levels of genes were calculated by the $2^{-\Delta\Delta Ct}$ method [26].

*2.4. Statistical Analysis*

Photosynthetic pigment contents, physiological indexes and qRT–PCR validation of DEGs of *I. difengpi* under drought stress were analysed statistically with the software Statistical Product and Service Solutions (SPSS) version 22.0 (IBM Inc., New York, NY, USA). Homogeneity of variance was used to check the data distribution before running an analysis of variance (ANOVA). ANOVAs followed by Duncan's post hoc tests were used to test the statistical significance of differences ($p < 0.05$) between different experimental treatments. The software edge R was used for gene differential expression analysis. Fisher's exact test was used to measure significantly enriched terms, and the multiple testing method's false discovery rate was used for *p*-value correction. The *p*-value threshold was ≤0.05. SigmaPlot 9.0 (SPSS Inc., Chicago, IL, USA), and Origin 8.0 was used for drawing.

**3. Results**

*3.1. Physiological Responses of I. difengpi to Drought Stress*

Chlorophyll (Chl) contents of *I. difengpi* decreased with increasing drought stress (Figure 2a,b,d), while the chlorophyll a/b ratio showed almost no change (Figure 2e). The carotenoid (Car) content was slightly increased under mild stress, but when drought stress was further aggravated, it was significantly reduced (Figure 2c). Compared with the control, the ratio of Car/Chl significantly increased under mild stress, but it did not significantly increase under severe stress (Figure 2f). These data indicate that drought stress reduced the chlorophyll content of *I. difengpi*, thus weakening its photosynthetic capacity. However, under mild stress, the species could increase the relative content of carotenoids in photosynthetic pigments to promote thermal dissipation and thus prevent the plants from photoinhibition.

Under drought conditions, plants regulate cell osmotic pressure by accumulating large amounts of free proline and soluble sugar so that they can adapt to drought stress. In this study, the proline and soluble sugar contents of *I. difengpi* grown under different soil moisture conditions were determined, and the results indicated that the proline and soluble sugar contents increased with increasing drought stress (Figure 3a,b). Drought stress also had a significant impact on SOD and POD activities and MDA content in *I. difengpi* (Duncan's post hoc tests following ANOVA: $p < 0.05$) (Figure 3c–e). The activities of SOD and POD increased as the degree of drought stress increased; the MDA content showed no significant change under mild drought stress, but it increased under severe drought stress. The above results indicate that *I. difengpi* can resist the adverse effects (such as membrane lipid peroxidation) of drought stress by accumulating osmoregulatory substances and

enhancing the activity of antioxidant enzymes. This measure is particularly effective under mild stress.

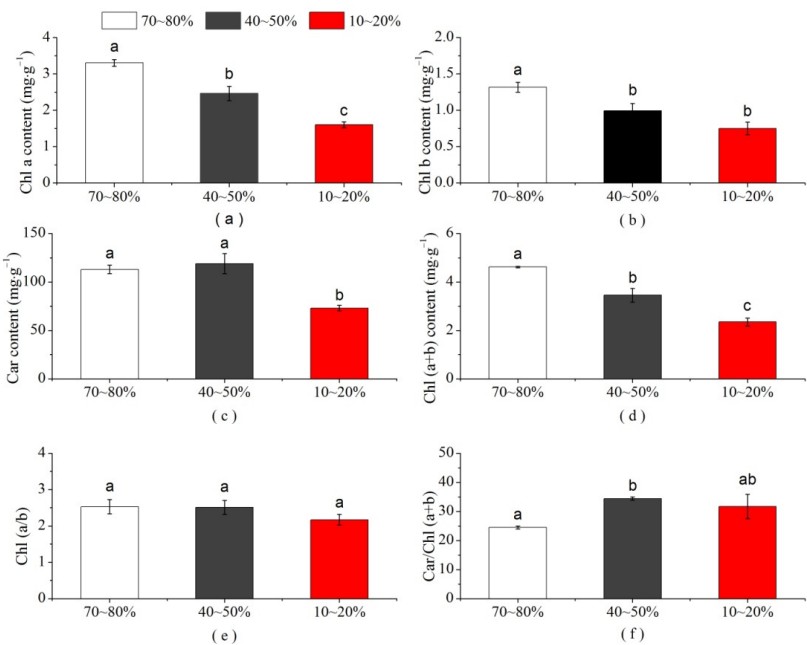

**Figure 2.** Photosynthetic pigment contents of *I. difengpi* under drought stress: (**a**) content of chlorophyll a, (**b**) content of chlorophyll b, (**c**) content of carotenoid, (**d**) content of chlorophyll (a + b), (**e**) ratio of chlorophyll a/b, (**f**) ratio of carotenoid/chlorophyll (a + b). Different letters above bars indicate significant differences among different light levels (Duncan's post hoc tests following ANOVA: $p < 0.05$), the error bars refer to standard deviations (means ± SDs).

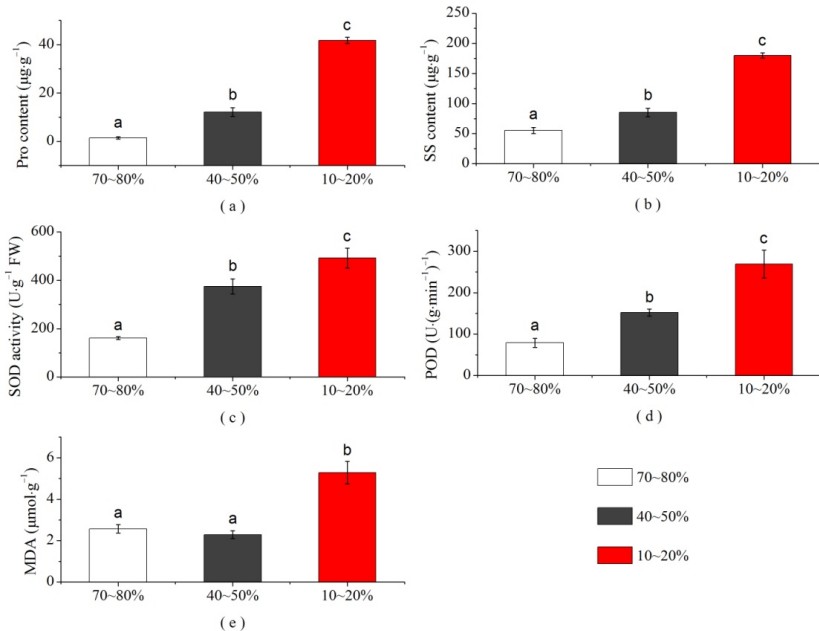

**Figure 3.** Physiological indexes of *I. difengpi* under drought stress: (**a**) proline content, (**b**) soluble sugar content, (**c**) superoxide dismutase activity, (**d**) peroxidase activity, and (**e**) malondialdehyde content. Different letters above bars indicate significant differences among different light levels (Duncan's post hoc tests following ANOVA: $p < 0.05$), the error bars refer to standard deviations (means ± SDs).

### 3.2. Quality Control and Assembly of Sequencing Data

Through the transcriptome sequencing of nine samples from the control (70~80%) and drought stress groups (40~50%, 10~20%), a total of 534,746,598 raw reads were generated, consisting of 80,746,736,298 bp (80.75 G). To ensure the accuracy of the subsequent bioinformatics analysis, the original sequencing data were filtered, and 519,672,982 clean reads were obtained, containing 76,650,655,479 bp (76.65 G, 94.93% of the original data). The Q20, Q30 and GC contents were approximately 97.33~97.92%, 92.68~93.98% and 46.00~48.48%, respectively, and the base error rate was less than 0.02%, indicating that the overall quality of the sequencing data was good (Table S1).

The clean reads were assembled using Trinity and yielded 77,964 unigenes containing 62,050,696 bp bases. These unigenes exhibited lengths of 201–16,837 bp, an average length of 795.89 bp and an N50 of 1294 bp. The length distribution of the assembled unigenes is shown in Figure S1.

### 3.3. Gene Function Annotation and DEG Analysis

The assembled unigenes were subjected to comparisons against the Nr, STRING, Pfam, SwissProt and KEGG databases, and the results showed that 39,383 (50.51%) unigenes were annotated in at least one database. Among these unigenes, 38,938 (49.94%), 13,493 (17.31%), 19,398 (24.88%), 22,823 (29.27%) and 15,198 (19.49%) were annotated in the Nr, STRING, Pfam, SwissProt and KEGG databases, respectively. The specific annotation information is shown in Table S2.

The expression levels of unigenes were expressed as FPKM values. Edge R software was used to perform differential expression analysis by comparing pairs of treatments (CK70~80 vs. XP40~50, CK70~80 vs. XP10~20, XP40~50 vs. XP10~20) (Table S3), and a total of 5343 unigenes showed significant differential expression in at least one comparison (Table S4). Compared with the control group (CK70~80), the number of DEGs in the drought stress groups increased with an increasing degree of drought, but fewer DEGs were identified in the XP40-50 vs. XP10-20 comparison (Table 1). In all comparisons, there were more downregulated genes than upregulated genes. Moreover, among these DEGs, 124 genes were common in the three comparisons, and with the exception of the four genes "c93445_g1", "c100967_g1", "c101528_g1" and "c95352_g1", the remaining 120 common DEGs showed either "consistent upregulation" or "consistent downregulation" (Figure S2, Table S4). This result suggests that a highly complex transcriptome is expressed when *I. difengpi* faces drought stress.

**Table 1.** Number of DEGs in each comparison.

| Comparisons | CK70-80 vs. XP40-50 | CK70-80 vs. XP10-20 | XP40-50 vs. XP10-20 |
|---|---|---|---|
| DEG numbers | 2489 | 4451 | 753 |
| Significantly up | 1137 | 1658 | 214 |
| Significantly down | 1352 | 2793 | 539 |

### 3.4. Classification of DEGs

To understand the molecular mechanism of the *I. difengpi* response to different degrees of drought stress, the 5343 DEGs were divided into seven clusters: early upregulated/downregulated genes, late upregulated/downregulated genes, continuously upregulated/downregulated genes and others (Table S5), according to their expression trends during the drought stress experiment. Among these clusters, the first six attracted our attention and were used to perform GO enrichment and KEGG pathway enrichment analysis (Figures S3 and S4, Tables S6 and S7).

Some early upregulated/downregulated genes identified in the CK70~80 vs. XP40~50 comparison did not present the same expression trend in the XP40~50 vs. XP10~20 comparison. There were 1113 early upregulated genes, among which the most significantly enriched GO terms were associated with stress responses (Figure S3A), including terms such as

response to stimulus (GO:0050896), response to stress (GO:0006950), response to chemical (GO:0042221) and response to oxygen-containing compound (GO:1901700). The number of early downregulated genes was 1256. This gene cluster exhibited the most significant enrichment in chloroplast/photosynthesis component- and enzymatic activity-associated GO terms (Figure S3B), such as thylakoid (GO:0009579), thylakoid part (GO:0044436), plastid part (GO:0044435), chloroplast part (GO:0044434) and catalytic activity (GO:0003824).

The late upregulated/downregulated genes included some genes that were upregulated/downregulated in the CK70~80 vs. XP10~20 comparison but showed no significant change in the CK70~80 vs. XP40~50 comparison. A total of 961 late upregulated genes were detected among all the DEGs, with the most significant enrichment in GO terms being related to the membrane system (Figure S3C), including terms such as the intrinsic component of membrane (GO:0031224), integral component of membrane (GO:0016021), endoplasmic reticulum (GO:0005783), membrane (GO:0016020) and endomembrane system (GO:0012505). A total of 1745 late downregulated genes were identified, and the GO terms with the most significant enrichment were photosystem II (GO:0009523), chloroplast thylakoid (GO:0009534), photosynthesis (GO:0015979), photosynthesis, and light reaction (GO:0019684) (Figure S3D).

The continuously upregulated/downregulated genes were the genes that were consistently upregulated or downregulated in all three comparisons. In this study, 24 continuously upregulated genes were identified among the DEGs. This category included the genes enriched in the GO terms involved in chitin metabolism, amino sugar metabolism, cell wall metabolism and glycolysis (Figure S3E). At the same time, 96 continuously downregulated genes were also screened from the DEGs. Many of these genes were significantly enriched in GO terms related to chloroplast/photosynthesis components and membrane systems (Figure S3F).

Moreover, KEGG pathway enrichment analysis was also performed with these DEGs, and the results are shown in Figure S4 and Table S7. Here, we found that the early upregulated genes showed significant enrichment in four pathways, which were "plant hormone signal transduction", "biosynthesis of secondary metabolites", "plant–pathogen interaction" and "biosynthesis of amino acids"(Figure S4A), while the early downregulated genes showed significant enrichment in 10 pathways, of which the five best-represented pathways were "plant hormone signal transduction", "biosynthesis of secondary metabolites", "chloroalkane and chloroalkene degradation", "flavonoid biosynthesis" and "drug metabolism-cytochrome P450" (Figure S4B). The late upregulated genes were only significantly enriched in the pathway "protein processing in endoplasmic reticulum", which was followed by "plant hormone signal transduction", "glycolysis/gluconeogenesis" and "biosynthesis of unsaturated fatty acids" (Figure S4C); simultaneously, the main enriched DEGs in late downregulated genes were involved in 15 pathways, such as "photosynthesis", "biosynthesis of secondary metabolites", "photosynthesis-antenna proteins" and "porphyrin and chlorophyll metabolism" (Figure S4D). In addition, the continuously upregulated genes only showed significant enrichment in "glycolysis/gluconeogenesis" (Figure S4E), while the continuously downregulated genes showed significant enrichment in 10 pathways, such as "glyoxylate and dicarboxylate metabolism", "carbon metabolism", "carbon fixation in photosynthetic organisms", "photosynthesis-antenna proteins" and "metabolic pathways" (Figure S4F).

*3.5. DEGs Related to Stress Responses in I. difengpi*

Through GO enrichment analysis, we found that the DEGs in the early upregulation cluster showed significant enrichment in multiple GO terms associated with stress responses. An in-depth analysis of these GO terms identified some genes involved in plant drought stress responses, including six heat shock proteins (HSP), three SNF1-related protein kinases 2 (SnRK2s), one homeobox leucine zipper family protein (HD-ZIP), two galactinol synthase (GOLS), and three ethylene-responsive transcription factor (EREBP) (Table S8). These genes are involved in the plant stress response, abscisic acid signalling

pathway, osmotic regulation or ethylene signalling pathway. In the later upregulated cluster, DEGs were significantly enriched in GO terms related to the membrane system, including five plasma membrane-type ATPases (P-ATPase) and 1 acyl-CoA desaturase (desC) (Table S8). These proteins were related to transmembrane transport, lipid metabolism and fatty acid desaturation. There were 24 DEGs classified as being continuously upregulated after exposure to drought stress, among which only five genes were included in the significantly enriched GO terms. These five genes contained three genes with detailed annotations, including two chitinases and one glyceraldehyde-3-phosphate dehydrogenase (GAPA, cytosolic-type) (Table S8), which were associated with the chitin catabolic process and glucose metabolic process.

Simultaneously, the downregulated DEGs in different clusters all had a significant enrichment in the GO terms related to photosynthesis (Figures 4 and 5, Table S8). In the early downregulated cluster, 12 DEGs attracted our attention, encoding three photosynthesis-antenna proteins (one LHCA4 protein and two LHCB4 proteins), five photosynthesis process-related proteins (two psaA proteins, one psbA protein and two psbP proteins), two chlorophyll synthesis enzyme proteins (magnesium-protoporphyrin IX monomethyl ester (oxidative) cyclase and magnesium-chelatase) and one carbon dioxide assimilation-related enzyme protein (ribulose bisphosphate carboxylase protein). DEGs derived from the late downregulation category also contained a series of genes related to photosynthesis, from which we identified 10 photosynthesis-antenna protein-related genes, 27 photosynthesis process-related genes, 4 chlorophyll synthesis-related genes and 2 carbon dioxide assimilation-related genes. These genes encoded proteins including LHCA1, LHCA2, LHCB1, LHCB2, psaB, psaF, psbO, psbP, ferredoxin I (petF), protochlorophyllide reductase, ribose-5-phosphate isomerase 3 and fructose-1,6-bisphosphatase, etc. In the continuously downregulated cluster, five photosynthesis-antenna protein-related genes, four photosynthesis process-related genes and four carbon dioxide assimilation-related genes were identified in this study, which encode proteins such as LHCA2, LHCB1, psaD, psbQ, glyceraldehyde 3-phosphate dehydrogenase (GAPA, chloroplastic-type) and 2 ribulose-1,5-bisphosphate carboxylase/oxygenase.

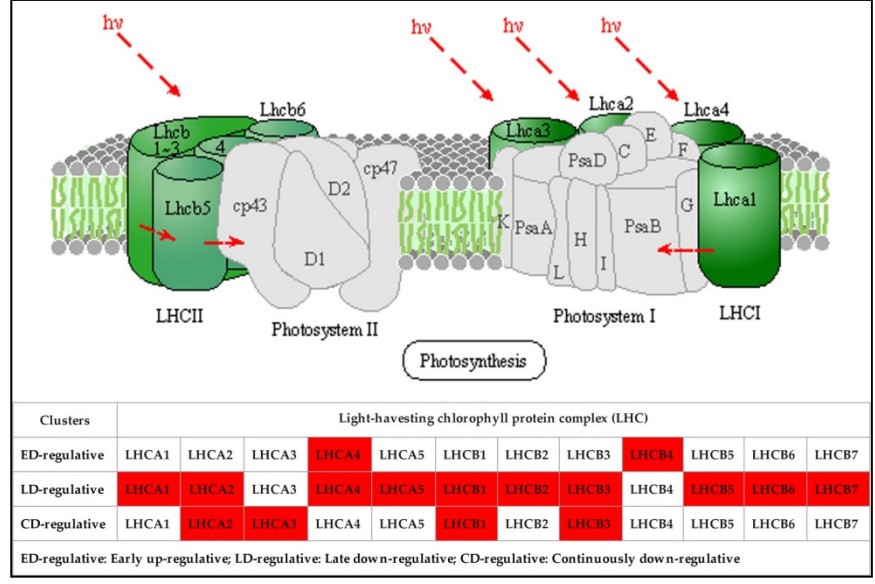

(**a**)

**Figure 4.** *Cont.*

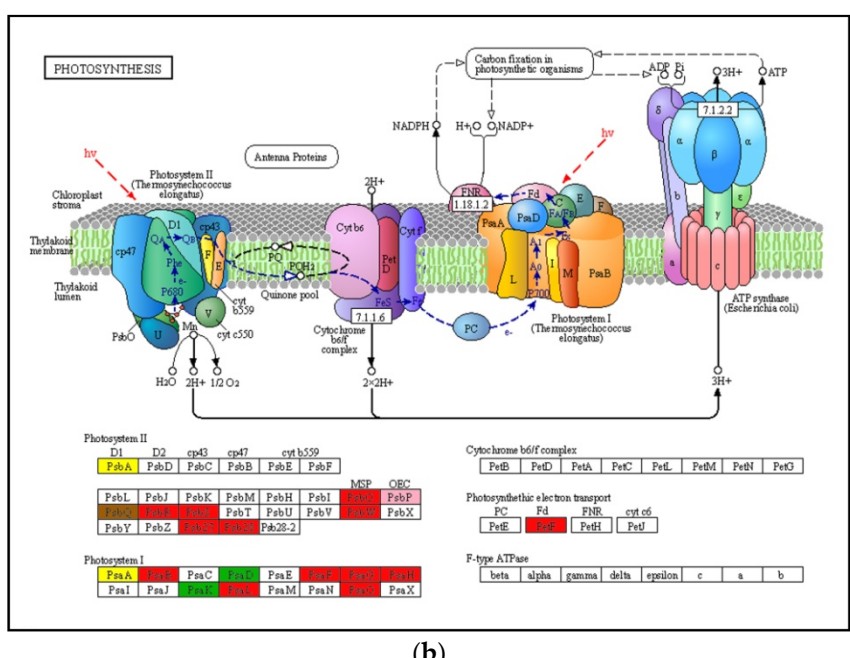

**(b)**

**Figure 4.** DEGs involved in photosynthesis in different downregulated clusters. (**a**) DEGs related to photosynthesis-antenna proteins; red symbols in different clusters indicate downregulated genes. (**b**) DEGs related to the photosynthesis process; yellow symbols indicate the genes derived from the early downregulated cluster; red symbols indicate the genes classified as the late downregulated cluster; green symbols indicate continuously downregulated genes; pink symbols indicate the genes derived from the early downregulated and late downregulated clusters; brown symbols indicate the genes derived from the late downregulated and continuous clusters.

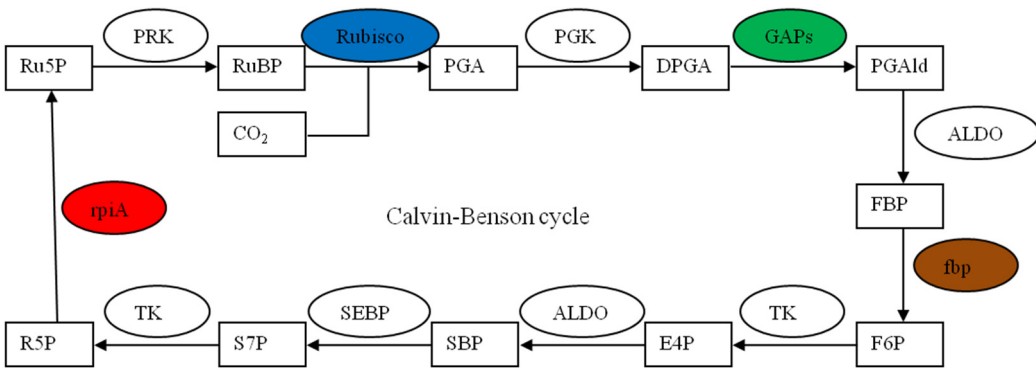

**Figure 5.** DEGs involved in carbon dioxide assimilation in different clusters. Intermediate metabolites in circulation: Ru5P, D-Ribulose 5-phosphate; RuBP, D-Ribulose 1,5-bisphosphate; PGA, 3-Phospho-D-glycerate; DPGA, 1,3-Bisphospho-D-glycerate; PGAld, D-Glyceraldehyde 3-phosphate; FBP, D-Fructose 1,6-bisphosphate; F6P, D-Fructose 6-phosphate; E4P, D-Erythrose 4-phosphate; SBP, D-Sedoheptulose 1,7-bisphosphate; S7P, D-Sedoheptulose 7-phosphate; R5P, D-Ribose 5-phosphate. Enzymes in circulation: PRK, 5-phosphoribulose kinase; Rubisco, ribulose-bisphosphate carboxylase; PGK, phosphoglycerate kinase; GAPs, glyceraldehyde 3-phosphate dehydrogenases (chloroplastic-type); ALDO, fructose-bisphosphate aldolase; fbp, fructose-1,6-bisphosphatase I; TK, transketolase; SEBP, sedoheptulose-1,7-bisphosphatase; rpiA, ribose 5-phosphate isomerase A. Red symbol indicates the genes classified as late downregulated cluster; green symbol indicates continuously downregulated genes; blue symbol indicates the genes derived from the early downregulated and continuously downregulated clusters; brown symbol indicates the genes derived from the late downregulated and continuous clusters.

Moreover, KEGG pathways related to the drought response were also identified. In the early upregulation cluster, "plant hormone signal transduction" was the most significantly enriched pathway (Figure 6, Table S9) and contained 19 DEGs encoding 8 abscisic acid-responsive proteins (phosphatase 2C proteins, PP2Cs; serine/threonine-protein kinase proteins, SNRK2s), 4 auxin-responsive proteins (Aux/IAA, SAUR, GH3), 2 gibberellin receptor proteins (GID1s), 4 two-component regulatory system proteins (A-ARRs, AHKs) and 1 cyclin D3 protein. In the late upregulation cluster, a series of DEGs encoding proteins related to protein folding and modification were identified in the only enriched pathway "protein processing in endoplasmic reticulum". These encoded proteins included four membrane-bound transcription factor proteins (S1P, S2P), one E3 ubiquitin protein ligase RIN2 protein, two protein disulfide-isomerase proteins (PDIs), three heat shock proteins (sHSP, HSP70, HSP90), two mannosyl-oligosaccharide 1,2-alpha-mannosidase (MAN1) proteins, two dolichyl-diphosphoooligosaccharide–protein glycosyltransferase proteins (OSTs) and one Protein OS-9 (OS-9) (Figure 7, Table S9). However, only one pyruvate kinase (PK) gene and one glyceraldehyde-3-phosphate dehydrogenase (GAPA, cytosolic-type) gene were significantly enriched in the "glycolysis/gluconeogenesis" pathway, which was the only KEGG pathway with a significant enrichment in the continuously upregulated category (Tables S7 and S9).

Among the ten significantly enriched pathways in the early downregulated cluster, "plant hormone signal transduction" was the most important pathway, containing eight genes related to the IAA signalling pathway, two genes related to the BR signalling pathway, three genes associated with ABA signal transduction and two JA signalling pathway genes (Figure 6, Table S9). In late/continuously downregulated clusters, multiple genes involved in plant photosynthesis were identified, including LHCAs, LHCBs, psas and psbs. Moreover, 16 genes involved in chlorophyll metabolism were also identified in late downregulated clusters (Table S9).

### 3.6. DEGs of Transcription Factors (TFs) under Drought Stress

In this study, a total of 244 TFs from 10 families showing differential expression under mild stress (XP40-50) or severe stress (XP10-20) were identified (Figure 8, Table S10). These TFs mainly belonged to the zinc finger protein (C2H2, C3H, C4, C5HC2 and CCCH), MYB, EREBP, NAC and WRKY families, which accounted for 88 (36.07%), 39 (15.98%), 26 (10.66%), 26 (10.66%), and 22 (9.02%) of the DEGs, respectively. Compared with the control, 78 and 52 TFs in the mild stress group were upregulated and downregulated, respectively. Among these TFs, the EREBP, NAC, bZIP, MYB and WRKY families exhibited more upregulated genes than downregulated genes, while the zinc finger protein (C2H2, C3H, C4, C5HC2 and CCCH) and bHLH families showed the opposite pattern. In the severe stress group, 94 and 106 TFs showed upregulation and downregulation, respectively, among which the EREBP, NAC, bZIP, MYB and TIFY families exhibited more upregulated genes, while the WRKY, zinc finger protein (C2H2, C3H, C4 and CCCH) and bHLH families exhibited more downregulated genes. Distribution analysis of the differentially expressed TFs in the six clusters, and the results are shown in Table 2. Through this result, we found that nearly all classes of TFs with more upregulated genes than downregulated genes in mild or severe stress also have more genes distributed in the early upregulated cluster than in other clusters, especially EREBP, NAC, bZIP and MYB. These results indicate that TFs that positively respond to drought stress usually begin to function at early or mild drought stress.

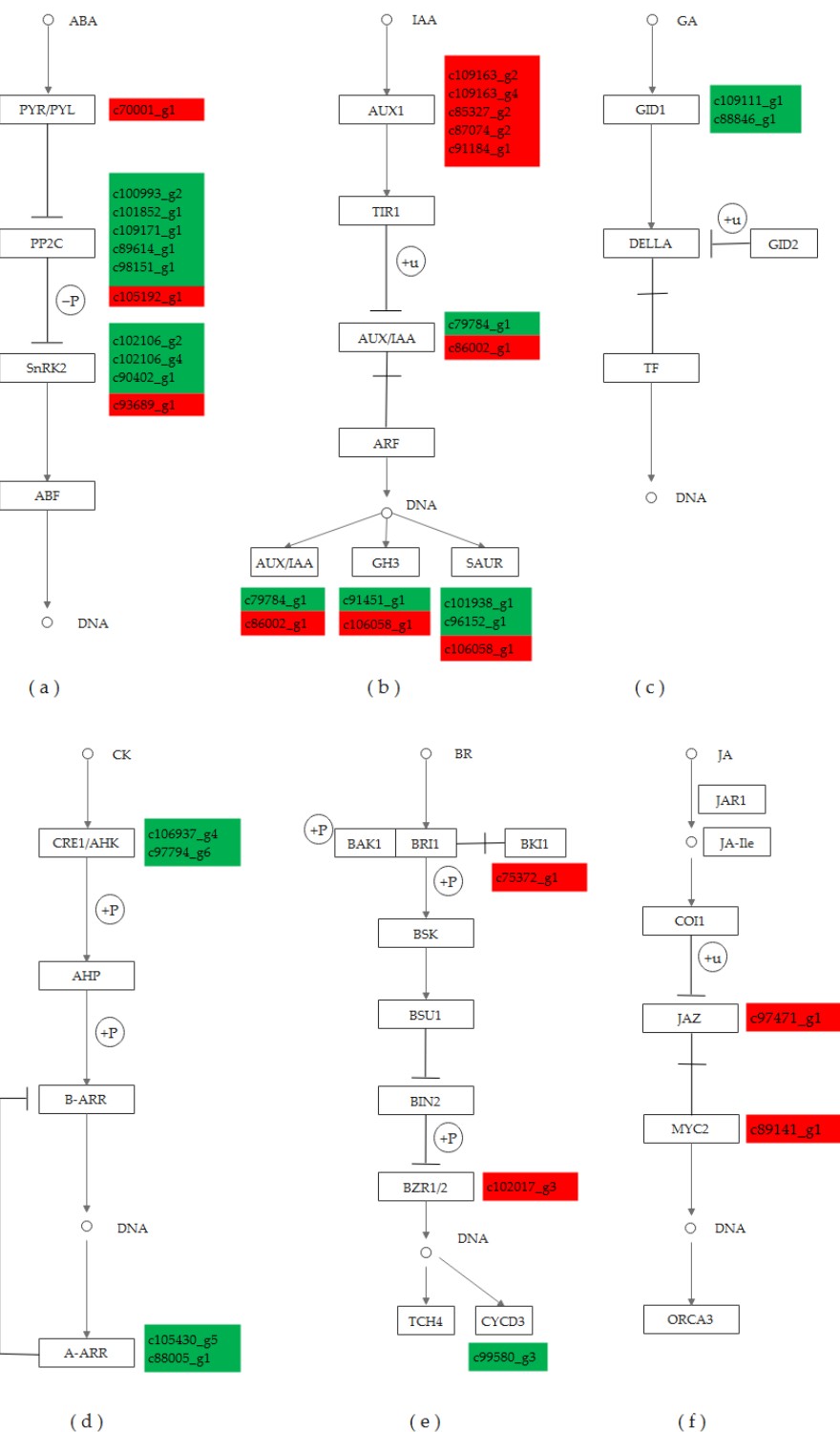

**Figure 6.** DEGs in the plant hormone signal transduction pathway under mild drought stress. (**a**) ABA signal transduction pathway; (**b**) IAA signal transduction pathway; (**c**) GA signal transduction pathway; (**d**) CK signal transduction pathway; (**e**) BR signal transduction pathway; (**f**) JA signal transduction pathway. Genes marked in green were upregulated, while those marked in red were downregulated.

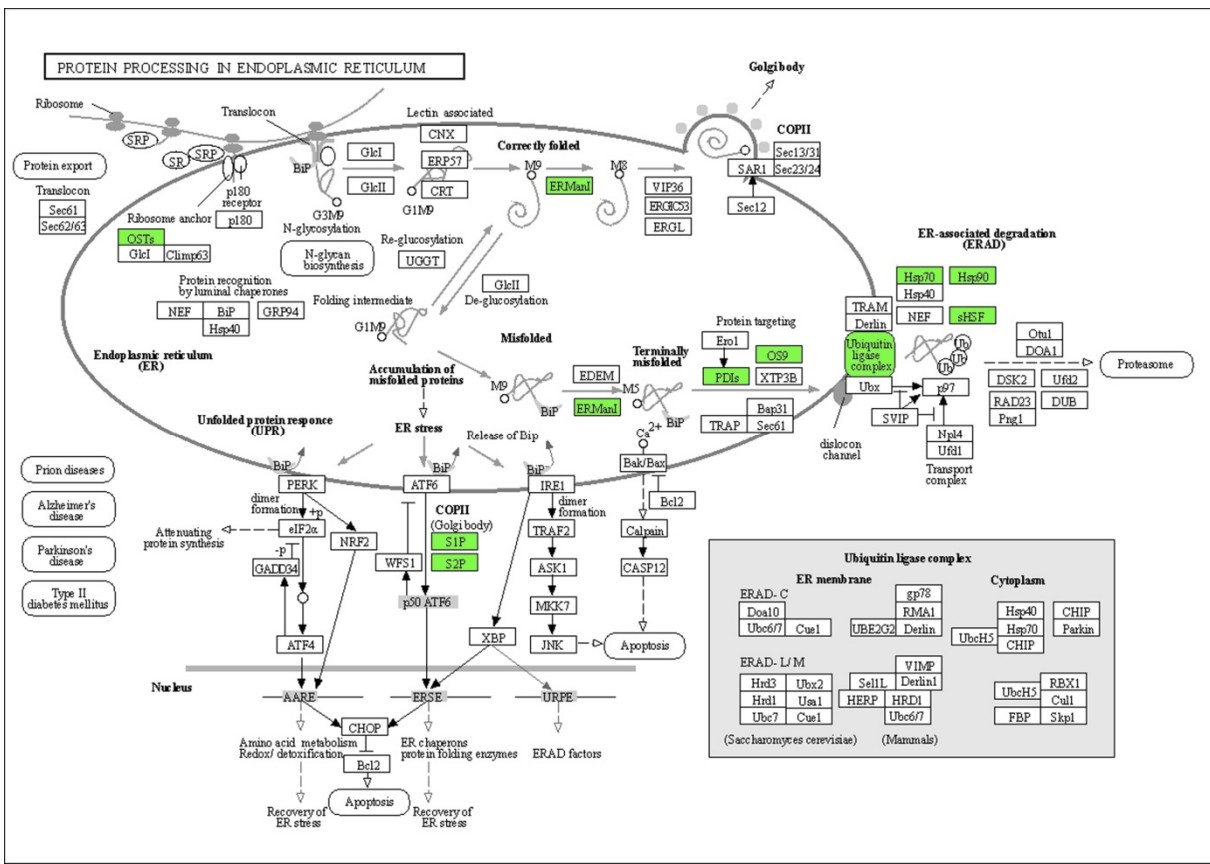

**Figure 7.** DEGs in protein processing in the endoplasmic reticulum pathway under severe drought stress. Light green symbols indicate that the genes were upregulated.

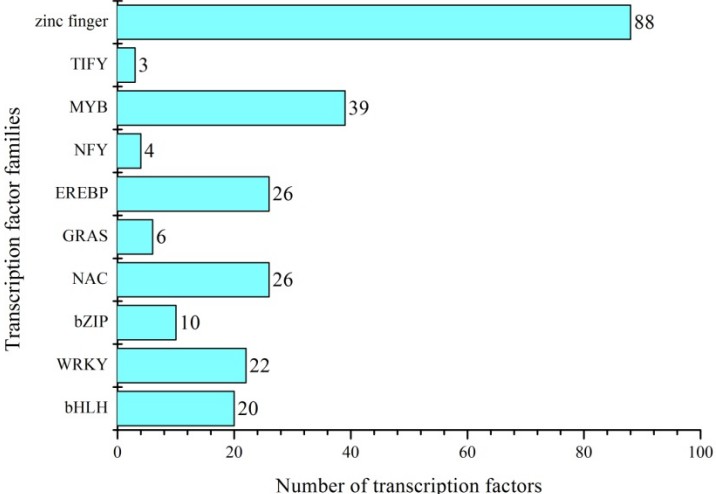

**Figure 8.** Transcription factor classification and number of DEGs associated with the response to drought stress. In total, 244 TFs from 10 families showing differential expression under light stress or severe stress were identified.

**Table 2.** Distribution of 244 differentially expressed transcription factors in different clusters.

| Transcription Factors | Number of Transcription Factors | | | | | |
|---|---|---|---|---|---|---|
| | Early Upregulated | Early Downregulated | Late Upregulated | Late Downregulated | Continuously Upregulated | Continuously Downregulated |
| zinc finger | 15 | 23 | 25 | 25 | 0 | 0 |
| TIFY | 1 | 1 | 1 | 0 | 0 | 0 |
| MYB | 14 | 7 | 9 | 9 | 0 | 0 |
| NFY | 0 | 0 | 2 | 2 | 0 | 0 |
| EREBP | 20 | 0 | 0 | 6 | 0 | 0 |
| GRAS | 2 | 2 | 1 | 1 | 0 | 0 |
| NAC | 10 | 5 | 3 | 7 | 1 | 0 |
| bZIP | 7 | 0 | 2 | 1 | 0 | 0 |
| WRKY | 8 | 6 | 1 | 7 | 0 | 0 |
| bHLH | 0 | 6 | 5 | 7 | 0 | 2 |

*3.7. qRT–PCR Validation of DEGs*

To assess the accuracy of the RNA-seq data and to further confirm the patterns of differential gene expression, we selected six DEGs that may be related to drought stress for qRT–PCR verification. The results showed that the expression levels and patterns of the genes selected from the transcriptome sequencing results were largely consistent with the results of the qRT–PCR experiments, indicating that the data obtained from the transcriptome sequencing were reliable (Figure 9a,b).

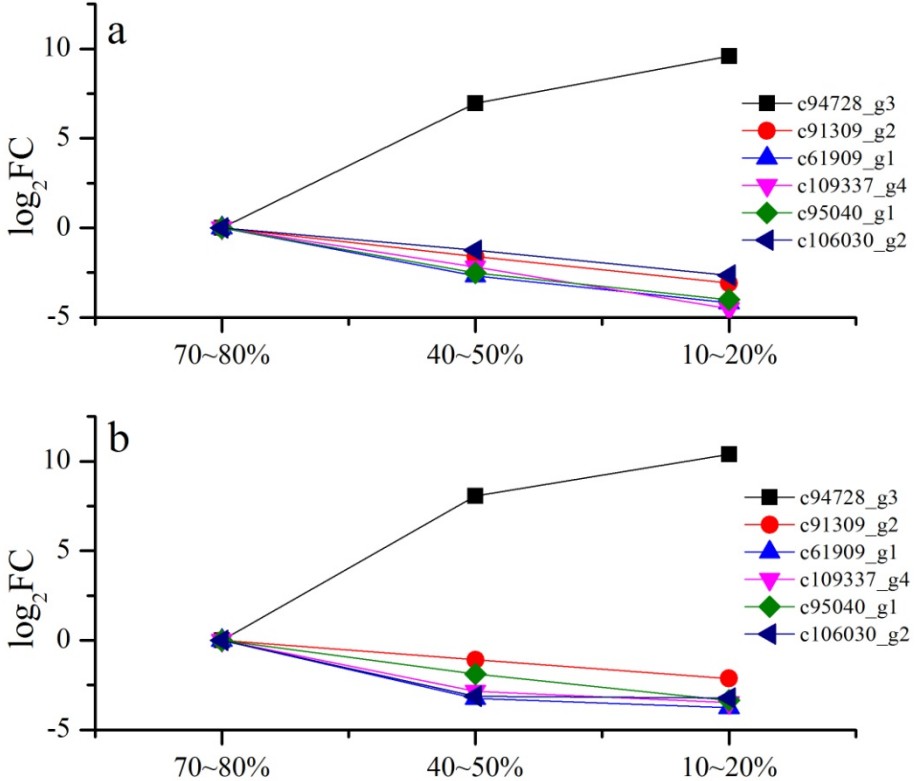

**Figure 9.** Expression patterns of the selected DEGs determined by RNA-Seq (**a**) and qRT–PCR (**b**). FC is the fold change in FPKM of a gene between the two samples.

**4. Discussion**

Water is indispensable for plant growth and development; however, water shortages have recently become a global concern [27], especially in karst areas and arid and semi-arid regions, which result in the plants in these areas being under frequent drought stress. Under

drought conditions, plants increase their tolerance to drought stress through a series of physiological and biochemical responses so that they can survive in arid environments [28]. These responses include repression of photosynthesis, the accumulation of osmoregulatory substances, and the enhancement of antioxidant enzyme activity [29,30]. In this study, we performed physiological and transcriptomic analyses of *I. difengpi* under drought conditions. Then, the differentially expressed genes in multiple signalling pathways that were potentially associated with drought were analysed with bioinformatics methods. The results provide a basis for the identification of drought resistance candidate genes in *I. difengpi* and provide informative clues for understanding its drought tolerance mechanism.

### 4.1. Impact of Drought Stress on Photosynthesis

Photosynthesis is a complex process that is mainly carried out in chloroplasts. When plants are in arid environments, the chlorophyll content will decrease due to the peroxidation of the chloroplast membrane, which damages the membrane system and blocks chlorophyll synthesis [31]. Additionally, the expression levels of genes encoding chloroplast development-related proteins and photosynthetic reaction centre multisubunit protein complex components will also be downregulated because of drought stress [32,33]. This series of physiological reactions will have an adverse effect on chloroplast development, chlorophyll synthesis and catabolism, and the molecular regulation of photosynthesis. In our study, the chlorophyll content of *I. difengpi* gradually decreased as the drought stress increased (Figure 2a,b,d), and similar results were reported by Ying et al. [34]. Based on GO enrichment analysis, we found that DEGs derived from different downregulated clusters all had a significant enrichment in the GO terms associated with photosynthesis. However, few PS I- and PS II-related proteins were downregulated in the early and continuously downregulated clusters. Instead, in the late downregulated cluster, multiple proteins related to photosynthesis were downregulated. In addition, we also identified 16 chlorophyll synthesis-related genes from the late downregulation category through KEGG enrichment analysis. These results indicated that severe drought stress caused great damage to the photosynthetic system of *I. difengpi,* which still maintains strong photosynthetic capacity under mild drought stress.

### 4.2. Hormone Metabolism and Signal Transduction in I. difengpi under Drought Stress

Plant growth and development are coregulated by various hormones. Under normal conditions, the content of these hormones is in a balanced state, but the balance is destroyed when plants are faced with drought stress conditions, which will affect their metabolism and growth [35]. In this study, KEGG enrichment analysis showed that DEGs were classified as early upregulated/downregulated categories with a significant enrichment in the "plant hormone signal transduction" pathway (Figure 6). The ABA and IAA signalling pathways, which had the largest number of differentially expressed genes, were the most active hormone signalling pathways in the early upregulated cluster and early downregulated cluster, respectively. ABA plays important roles in the adaptive response of plants to various abiotic stresses, and the mechanism has been extensively reported in the model plant *Arabidopsis* [36]. Pyracbactin resistance/pyracbactin resistance-like proteins (PYR/PYL) are soluble ABA receptors, and the PP2C-SnRK2 complex is the central regulator of the ABA signalling pathway. Their physiological functions in the ABA signalling pathway are vital [37]. In our results, eight out of eleven DEGs encoding proteins PYR/PYL, PP2Cs and SnRK2s in mild stress were upregulated (Figure 6a). This result suggests that ABA signalling was activated in the mild drought stress, which subsequently mediates and channelizes stomatal closure and stress-responsive gene expression. The most important role of auxin in plants is to promote growth and development. Previous studies have demonstrated that AUX1, an auxin influx carrier, is involved in regulating key plant processes, including lateral root initiation, leaf morphogenesis and female gametophyte development [38–40]. *AUX/IAA, GH3* and *SAUR* genes are three major classes of auxin-responsive genes, and the expression of these genes in pea, *Arabidopsis* and soybean

is rapidly induced by auxin [41–43]. In this study, most of the DEGs encoding AUX/IAA, GH3 and SAUR proteins in the IAA signalling pathway were downregulated, and five *AUX1* genes were all downregulated under mild drought stress (Figure 6b), indicating that *I. difengpi* may respond to mild drought stress by inhibiting the auxin signalling pathway and then slowing growth and development. Based on the results shown in Figure 6c,d, the DEGs involved in the CK and GA signalling pathways, including two *CRE1/AHK* genes, two *ARR-A* genes and two *gibberellin-insensitive dwarf1* genes (*GID1*), were all upregulated. Previous studies have shown that CK could increase drought tolerance through the coordinated regulation of carbon and nitrogen assimilation in rice [44]. Although CRE1/AHK proteins (a receptor of CK) play a negative regulatory role in plant drought resistance, they may be important in mediating the input of cytokinin into the salt stress response pathway [45]. Two-component response regulator A-ARR family proteins (A-ARRs) are partially redundant negative regulators of cytokinin signalling [46]. A previous study suggested that overexpression of *ARR5* in *Arabidopsis* showed ABA hypersensitivity and drought tolerance [47]. In another instance, the expression of the *MsGID1b* gene in *Medicago sativa* could be upregulated under NaCl and PEG treatments, indicating that the *MsGID1b* gene might participate in the response to abiotic stress of the species [48]. In addition, downregulated *BKI1*, *BZR1/2*, *JAZ* and *MYC2* genes in the Br and JA signalling pathways were identified. BKI1 is a negative regulator of Br signalling in the absence of Br [49]. However, when Br is present, it can play a dual role by antagonising a subset of 14-3-3 proteins and enhancing the BRI1 EMS SUPPRESSOR1 (BES1)/BRASSINAZOLE RESISTANT 1 (BZR1), a positive regulator of Br signalling, accumulated in the nucleus to activate BR-responsive gene expression [50,51]. In *Arabidopsis*, mutation of the *BKI1* gene makes plants sensitive to salt, whereas transgenic plants overexpressing *BKI1* show a salt-tolerant phenotype [50]. Jasmonic acid is a hormone that can improve plant resistance to stress. JAZ family proteins act as JA coreceptors and transcriptional repressors in JA signalling, and MYC2 acts as both an activator and repressor of distinct JA-responsive gene expression [52,53]. The expression of the genes related to the Br and JA signalling pathways was downregulated in our study, suggesting that negative regulation occurred in these two pathways in *I. difengpi* under drought stress.

*4.3. Impact of Drought Stress on the Membrane System*

The existence of the plasma membrane makes the cell have a relatively stable intracellular environment. In this study, the MDA content was significantly increased in *I. difengpi* under severe drought stress (Figure 3e). MDA content is an important index to evaluate the degree of membrane lipid peroxidation [54]. When plants are under abiotic stress, membrane lipid peroxidation causes an increase in membrane permeability and leads to electrolysis leakage and a drop in water potential [55]. The accumulation of MDA in *I. difengpi* indicated that the membrane system of the species was damaged under severe drought stress. Reactive oxygen species (ROS) are the main cause of membrane lipid peroxidation [56]. To reduce damage, plants scavenge ROS via the antioxidant enzyme system. In our results, the antioxidant enzyme (SOD, POD) activities (Figure 3c,d) in *I. difengpi* were all significantly increased under both mild and severe drought stress, which may mean that antioxidant enzymes play an important role in scavenging ROS under drought conditions, and similar results were also reported by *Liu* et al. [57]. In addition to alleviating the toxic effect of ROS on the membrane system by enhancing the activity of antioxidant enzymes, *I. difengpi* also resists membrane lipid peroxidation by enhancing the expression of fatty acid desaturases. In our experiment, an acyl-CoA desaturase (desC) gene was identified from the late upregulated cluster. Interestingly, a previous study reported that tobacco plants transformed with the *desC* gene more efficiently resisted the accumulation of reactive oxygen species and reduced the rate of lipid peroxidation [58].

In addition, the plasma membrane also provides a large number of attachment sites for enzymes where active transport and biochemical reactions can occur. P-ATPases are the major enzyme proteins attached to the plasma membrane, including five major subfamilies

(heavy-metal ATPases, $Ca^{2+}$-ATPases, $H^+$-ATPases, putative aminophospholipid ATPases and a branch with unknown specificity) [59]. In our study, one $H^+$-ATPase, two $Ca^{2+}$-ATPases and two aminophospholipid ATPases were identified in the late upregulated cluster (Table S8). Plasma membrane $H^+$-ATPases (PM $H^+$-ATPase) are proton pumps that can maintain ion homeostasis in cells by nutrient uptake, intracellular pH regulation and an increase in major osmolyte biosynthesis [60,61]. Plasma membrane $Ca^{2+}$-ATPases are calcium pumps that export $Ca^{2+}$ from the cytosol to the extracellular environment and thus maintain overall $Ca^{2+}$ homoeostasis and provide local control of intracellular $Ca^{2+}$ signalling [62], while aminophospholipid ATPases are enzymes that catalyse membrane flipping, driving the active transport of phospholipids from exoplasmic or luminal leaflets to cytosolic leaflets [63]. Several previous studies have demonstrated that these P-ATPases are involved in plant responses to abiotic stresses. For instance, *Avena sativa* adapts to drought stress by early activation of the root hair cell PM $H^+$-ATPase and then triggers the increased biosynthesis of major osmolytes, which in turn leads to the upregulation of the water maintenance system [61]. In *Oryza sativa*, the transcript level of the P-type IIB $Ca^{2+}$-ATPase gene *OsACA6* was enhanced in response to salt, drought, abscisic acid and heat, which then promoted stress tolerance in plants by ROS scavenging and enhancing the expression of stress-responsive genes [64]. Simultaneously, another study confirmed that ALA6, a P4-type ATPase responsible for flipping and stabilising asymmetric phospholipids in membrane systems, is involved in heat stress responses in *Arabidopsis thaliana* [65]. The increased expression of P-ATPases in *I. difengpi* indicates the enhancement of the synthesis and transport of major osmolytes, as well as the improvement of drought stress signal transmission in the species under severe drought stress.

*4.4. Analysis of Protein Processing in Endoplasmic Reticulum*

The endoplasmic reticulum (ER) is the main site for posttranslational processing, modification and folding of nascent peptides. After the nascent peptides are correctly folded, they will be transported to the golgibody. However, misfolded peptides are retained, and if misfolded persistently, they are selectively transported to the cytosol for degradation by proteasomes [66,67]. Before degradation, misfolded peptides are frequently ubiquitylated and then escorted to proteasomes by polyubiquitin-binding proteins [67]. Our results showed that severe drought stress significantly induced the "protein processing in endoplasmic reticulum" pathway in *I. difengpi*, and several genes related to ER-associated degradation biological processes were identified from the late upregulated cluster (Figure 7). These results suggest that *I. difengpi* enhanced the rate of misfolded protein degradation, which was produced by mass as a result of drought stress [68], to maintain cell homeostasis under stress. However, when misfolded proteins accumulate excessively and cannot be scavenged from the ER in a timely manner, the signal will be sensed by specific sensor proteins in the ER membrane [68], and ER stress will result. Subsequently, ATF6, an ER membrane-bound transcription factor, is hydrolysed and liberates the N-terminal cytosolic fragment ATF6(N) under the induction of ER stress and the catalysis of two proteases, S1P and S2P (site-1 and site-2 proteases) [69,70]. ATF6(N) then moves into the nucleus to activate unfolded protein response (UPR) target genes, which results in an increase in ER protein folding capacity and in turn mitigates ER stress [69]. In this study, two genes encoding S1P and S2P proteins were identified in *I. difengpi* under severe drought stress. Coincidentally, a previous study also claimed that ATF6 processing was blocked completely in cells lacking S2P and partially in cells lacking S1P [71]. The above results suggested that *I. difengpi* strengthens the quality monitoring of protein folding in the endoplasmic reticulum to maintain intracellular homeostasis under severe drought stress.

*4.5. Analysis of the Function of Enzymes Related to Chitin Catabolism and Glycolysis/Gluconeogenesis*

Chitinase is a glycoside hydrolase responsible for catalysing the hydrolysis of chitin, which is found in animals, plants and microorganisms. In plants, the expression of chiti-

nase is induced by biotic or abiotic stress and plays an important role in the response to stress [72]. Based on GO enrichment analysis, several GO terms related to chitin metabolism were identified in the continuously upregulated category. Two genes encoding class IV chitinase proteins were enriched in these GO terms. Interestingly, previous research also showed that most chitinases were upregulated under osmotic stress in Ammopiptanthus nanus leaves, among which two class IV chitinases were included [73]. Therefore, we speculated that class IV chitinase has a vital role in the response to different drought stresses in *I. difengpi*, but the mechanism needs to be further researched. KEGG pathway enrichment analysis indicated that "glycolysis/gluconeogenesis" was the only pathway with a significant enrichment in the continuously upregulated category, containing a pyruvate kinase (PK) gene and a glyceraldehyde-3-phosphate dehydrogenase (GAPDH) gene, and similar results were also reported in maize [74]. PK is the rate-limiting enzyme in the glycolytic pathway, catalysing the conversion of phosphoenolpyruvate and ADP to ATP and pyruvate. When plants are under drought stress, the reduction in stomatal conductance reduces oxygen uptake, which eventually leads to the weakening of the tricarboxylic acid cycle (TCA) and a decrease in ATP and NADH production. Upregulation of pyruvate kinase expression promotes the glycolytic pathway in plants and provides ATP and NADH for physiological and biochemical reactions under drought stress. Therefore, we suggest that the increase in PK expression may be an adaptive mechanism for *I. difengpi* to cope with drought stress through glycolysis. GAPDHs are present in the cytoplasm and chloroplasts of plants and are involved in glycolysis/gluconeogenesis and the Calvin cycle [75]. In rice, overexpression of *OsGAPC3*, a gene encoding cytosolic glyceraldehyde-3-phosphate dehydrogenase protein, increases the $H_2O_2$-scavenging ability of transgenic plants and alleviates oxidative stress [76]. Based on the expression pattern of the *GAPDH* gene in this study, we hypothesised that the GAPDH enzyme alleviates ROS-induced cell damage in *I. difengpi* under drought stress. In addition, the increase in PK and GAPDH expression also has a positive impact on carbon partitioning diverted from starch synthesis towards glycolysis, thereby providing more glycolytic intermediates and energy for the synthesis of other compounds related to drought tolerance [77,78].

*4.6. Response of TFs to Drought Stress*

TFs, also known as trans-acting factors, are a type of protein factor that can recognise and bind the core sequences of various cis-acting elements in DNA and thereby regulate the transcription frequency of target genes. When plants are subjected to biotic or abiotic stress, TFs are activated and interact with cis-acting elements, which then activate the RNA polymerase II transcription complex to initiate the expression of specific genes [79]. This study focused on the differentially expressed TFs in *I. difengpi* under drought stress, and a total of 244 TFs from 10 families were identified, including zinc finger proteins, MYB and NAC (Figure 8). Among these differentially expressed TFs, numerous TFs derived from the MYB, NAC, AP2/EREBP and bZIP families were upregulated.

There were 23 upregulated genes and 16 downregulated genes in the MYB family. This type of transcription factor contains a conserved MYB domain, and according to the number of adjacent repeats in the MYB domain, MYB factors can be divided into three subfamilies: MYB1R, R2R3-MYB, and MYB3R [80]. In *Arabidopsis*, Nakabayashi et al. [81] overexpressed the flavonol synthesis regulator gene *MYB12* under drought stress, and the results suggested a positive correlation between flavonoid accumulation and the prevention of water loss. In contrast, *AtMYB60* is an *R2R3-MYB* gene involved in the regulation of stomatal movement in *Arabidopsis*, and a null mutation in *AtMYB60* results in the constitutive reduction in the stomatal opening and in decreased wilting under drought stress conditions [82]. This indicates that the regulation of plant drought stress involving MYB family TFs is extremely complex. NAC, a plant-specific transcriptional regulatory factor, was first isolated from petunia in 1996 [83] and then discovered in *Arabidopsis*, rice, wheat and other species. Previous studies have shown that NAC TFs play important roles in plant growth, development and stress defence. In rice, the overexpression

of the stress-responsive gene *SNAC1* enhances drought resistance and salt tolerance in transgenic plants [84]. In this study, a total of 26 NAC TFs that were differentially expressed under mild or severe drought stress were identified, which included 11 upregulated TFs and 5 downregulated TFs in the mild stress groups as well as 10 upregulated TFs and 9 downregulated TFs in the severe stress groups. This result implied that most of the NAC TFs contributed to enhancing the drought tolerance of *I. difengpi*. AP2/EREBP TFs include five subfamilies, AP2, ERF, DREB, RAV and Soloist, and play roles in various processes throughout the plant life cycle, such as floral organ identity determination, leaf epidermal cell identity regulation, and the mechanisms by which plants respond to various biotic and environmental stresses [85,86]. In *Arabidopsis* and rice, the overexpression of a rice *OsDREB1F* gene makes transgenic plants more salt-, drought-, and low-temperature tolerant [87]. In this study, the 20 DEGs encoding AP2/EREBP TFs were all upregulated in the mild stress group, while there were 14 upregulated and 6 downregulated DEGs in the severe stress group. Among these DEGs, *c92025_g1* was found to show homology to *OsDREB1F*, and its expression was upregulated under both mild and severe stress. bZIP family TFs are often involved in ABA-regulated gene expression. For example, *HY5* and *ABI5* in *Arabidopsis* are not only involved in the signalling pathway activated by ABA during seed germination and seedling development but also required for osmotic stress tolerance in seedlings [88]. On the basis of transcriptome sequencing, two genes (gene IDs: *c101476_g1* and *c101823_g3*) with homology to *HY5* and *ABI5* were identified, but their expression patterns under drought stress were completely different. The expression of *c101476_g1* was upregulated under mild stress but showed no difference under severe stress compared with that in the control, while the expression of *c101823_g3* was downregulated under severe stress but did not show differential expression under mild stress.

In addition, our research also identified TFs from several other families. The expression patterns of these TFs under drought conditions were quite different, which may be related to the difference in the metabolic pathways in which they are involved. The transcriptome-wide identification of TFs not only helps to study the transcriptional control of many aspects of physiological characteristics in *I. difengpi* under drought stress, such as hormone response, signal transduction and secondary metabolite synthesis but also helps to identify the target genes of miRNA to construct regulatory networks at the posttranscriptional level. In future research, we will focus on those TFs that were upregulated under drought stress.

## 5. Conclusions

Karst areas are ecologically fragile areas. *I. difengpi*, a rare, endangered plant unique to these areas, has evolved an extremely strong tolerance to drought conditions. Therefore, *I. difengpi* can be a candidate tree for ecological restoration in karst areas, which not only benefits the ecological environment in karst areas but also protects the germplasm resources of the species. The results of this research focus on the physiological changes and molecular mechanisms of *I. difengpi* in response to drought stress. Our findings showed that the chlorophyll contents of *I. difengpi* decreased under drought stress, while the physiological indexes, such as Pro, SS and MDA contents and SOD and POD activities, all increased. Bioinformatics analysis revealed that *I. difengpi* responds to mild drought stress through hormone signal transduction and osmotic regulation. This species responds to severe drought by regulating intracellular osmotic pressure, relieving membrane lipid peroxidation and strengthening the quality monitoring of protein folding. Simultaneously, the chitin catabolism and glycolysis/gluconeogenesis pathways are also important to ensure that *I. difengpi* survives drought stress. In addition, numerous TFs related to drought stress were also identified in the present study. These results provide a preliminary understanding of the molecular mechanism of the *I. difengpi* response to drought stress. In the future, we will conduct multi-omics analysis combined with the genome and metabolome to find the core regulatory relationship in the drought resistance mechanism of this species. At the same time, rehydration experiments of *I. difengpi* under drought stress will also be

performed. These results will provide theoretical guidance for the application of the species in ecological restoration in karst areas.

**Supplementary Materials:** Supplementary materials can be found at https://www.mdpi.com/article/10.3390/su14127479/s1, Figure S1: Length distribution of assembled unigenes, Figure S2: Venn diagrams of DEGs across all comparisons, Figure S3: GO enrichment analysis of different clusters of DEGs, Figure S4: KEGG pathway enrichment analysis of different clusters DEGs, Table S1: Summary of the sequencing, Table S2: Annotation information of unigenes, Table S3: Differential expression analysis by comparing pairs of treatments, Table S4: Unigenes showing significant differential expression in at least one comparison, Table S5: Classification of DEGs, Table S6: GO enrichment analysis of different clusters DEGs, Table S7: KEGG pathway enrichment analysis of different clusters DEGs, Table S8: Drought Stress response related DEGs identified through GO enrichment, Table S9: Drought Stress response related DEGs identified through KEGG enrichment, Table S10: Differentially expressed transcription factors under drought stress.

**Author Contributions:** B.L., H.L. and H.T. conceived and designed the experiments. B.L., H.L. and M.W. performed the experiments. B.L., H.L., C.W., X.H., M.W. and H.T. analysed the data. X.W., M.W. and H.T. contributed reagents/materials/analysis tools. B.L. and H.L. wrote the paper. All authors have read and agreed to the published version of the manuscript.

**Funding:** This research was funded by the National Natural Science Foundation of China, grant number 32160093; the Project for Key Research and Development in Guangxi, grant number AB21220024; the Basic Research Fund of Guangxi Academy of Sciences, grant number CQZ-C-1901; the Guangxi Science and Technology Bases and Talent Project, grant number AD20297049; the Science and Technology Major Project of Guangxi, grant number AA18118015; the Project for Fundamental Research of Guangxi Institute of Botany, grant number 20001; the Project for Guilin Innovation Platform and Talent Plan, grant number 20210102-3; Botanic Gardens Conservation International (BGCI), grant number BGCI 30412.

**Institutional Review Board Statement:** Not applicable.

**Informed Consent Statement:** Not applicable.

**Data Availability Statement:** All data are available in the manuscript.

**Conflicts of Interest:** The authors declare no conflict of interest.

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
