# Peer review of "Physiological and Transcriptomic Responses of Illicium difengpi to Drought Stress"

_sustainability, doi:10.3390/su14127479_

Round 1
Reviewer 1 Report
The paper is interesting and well structured, but there is room for improvement.
I have provided many detailed comments in the attached pdf.
My major concern is about the lack of information about the statistical approach used for bioinformatics analyses. These should be provided in the section dedicated to Statistical analyses.
There are many sentences that must be revised and inconsistences that must be fixed. Some details must be provided at places. I have highlighted all parts that must revised/corrected/implemented in the attached pdf.

Author Response
Dear Editors and reviewers:
Thank you for your letter and for the reviewers’ comments concerning our manuscript entitled “Physiological and Transcriptomic Responses of Illicium difengpi to Drought Stress” (ID: sustainability-1713088). those comments are all valuable and very helpful for revising and improving our manuscript, as well as the important guiding significance to our researches. We have studied comments carefully and have made correction which we hope meet with approval. Revised portion are marked in yellow in the paper. The main corrections in the paper and the responds to the reviewers’ comments are as follows:
reviewer 1:
Comment 1: The paper is interesting and well structured, but there is room for improvement. I have provided many detailed comments in the attached pdf.
Response 1: Thank you very much for your detailed comments, which are very helpful to improve our manuscript. We have corrected most of the errors you pointed out in the manuscript according to your comments, and all the revision were marked in yellow in the revised manuscript. But there are several points we need to give explanations as follows:
- In “Abstract”, the expression “mild or early drought stress” means that, drought stress is a continuous process from mild stress to severe stress, in which mild stress is usually occurred in the early drought stage (early drought stress), while severe stress is occurred in the late drought stage (late drought stress). Therefore, we divided the differentially expressed genes (DEGs) into early upregulated/downregulated genes, late upregulated/downregulated genes, continuously upregulated/downregulated genes and others according to their expression trends under different drought stress. Genes that are differentially expressed under mild/severe drought stress means that they are differentially expressed under early/late drought stress. However, since the terms "early drought stress" and "late drought stress" are rarely mentioned in the “Materials and Methods” and “Results” sections, the use of which in the “Abstract” and “Discussion” will leave the reader with misunderstanding.. For consistency, we have removed the term the terms "early drought stress" and "late drought stress" from the manuscript.
- In “3.4. Classification of DEGs”, your comment for the sentence on line 298-304 (original manuscript) reads "This sentence is missing something", but we are not able to find the missing part. We think you may be referring to the fact that there are six clusters of genes in the Figure S2 and Table S7, and we describe only one of them in this sentence. The occurrence of this misunderstanding may be due to the previous sentence “Moreover, KEGG pathway enrichment analysis was also performed with these DEGs 297 (Figure S2, Table S7)”. In order to make the expression of this sentence clearer, we have made a revision in the revised manuscript which is marked in yellow.
- In “3.5. DEGs related to stress responses in Illicium difengpi”, your comment for the sentence on line 379-382 (original manuscript) reads “Something is lacking in this sentence”, which we think you may be mean the sentence the sentence structure is incomplete. Therefore, we make adjustments to the sentence structure, which you can see at the yellow mark in the revised manuscript.
- In “4.2. Hormone Metabolism and Signal Transduction in Illicium difengpi under Drought Stress”, the sentence “eight out of eleven DEGs encoding proteins PYR/PYL, PP2Cs and SnRK2s in mild stress were upregulated (Figure 8 a)” which on line 492-493 means that, a total of eleven DEGs related ABA signal were identified in this study, among them eight genes which encoding proteins PYR/PYL, PP2Cs and SnRK2s were upregulated (Genes with different ID number may encode the same protein).
- In “Supplementary Materials”, the meaning of the two p-level columns in Tables 6-7 were explained as follow: during the GO enrichment/KEGG pathway enrichment analysis, Fisher's precision probability test was used to examine the significant enrichment of GO terms and KEGG pathways, according which p value was calculated as follow formula:
p=(aa+b)(cc+d)/(na+c)=(a+b)!(c+d)!(a+c)!(b+d)!/a!b!c!d!n!
the meanings of the different letters are as follows:
|
|
The number of DEGs |
The number of non-DEGs |
Total |
|
GO term / KEGG pathway A |
a |
b |
a+b |
|
Others GO terms / KEGG pathways |
c |
d |
c+d |
|
Total |
a+c |
b+d |
a+b+c+d (=n) |
In order to reduce the false positive rate, p-values were adjusted using 4 multiple testing methods (Bonferroni, Holm, Sidak and false discovery rate), which and then obtained p-adjust. The p-adjust (≤0.05) were used to determine whether the GO terms/KEGG pathways with significant enrichment.
Comment 2: My major concern is about the lack of information about the statistical approach used for bioinformatics analyses. These should be provided in the section dedicated to Statistical analyses.
Response 2: The information about the statistical approach used for bioinformatics analyses had been added in our revised manuscript, and you can see it in the “2.4 Statistical analysis” section (Software edge R was used for gene differential expression analysis……Origin 8.0 were used for drawing).
Comment 3: There are many sentences that must be revised and inconsistences that must be fixed. Some details must be provided at places. I have highlighted all parts that must revised/corrected/implemented in the attached pdf.
Response 3: We have corrected most of the errors you pointed out in the manuscript according to your comments, and the revision instructions can be found in Response 1.
Reviewer 2 Report
The ms sustainability-1713088 with the title of Physiological and Transcriptomic Responses of Illicium difengpi to Drought Stress investigated the physiological index determination and transcriptome sequencing experiments in biennial Illicium difengpi seedlings grown under different soil moisture conditions (70~80%, 40~50% and 10~20%) to elucidate the molecular mechanisms of the response to drought stress in Illicium difengpi. The ms have to be significantly improved before it can go for father process.
Abstract:
The authors should make it shorter by reducing the text in the first two sentences (background) and mention only the key results.
The variations ((70~80%, 40~50% and 10~20%) or the range for the soil moisture conditions within each treatment is big (10%) and this can cause a mistake in the obtained results. 10% range is high error, and the authors should have made it accurate to get accurate results.
The treatments are very simple and I was looking to see that the authors used some treatments to mitigate the water or drought stress. But, this was not found.
Introduction:
The authors wrote long text about the Illicium difengpi, but they did not write deeply about the drought stress. They should write one more paragraph of few sentences about the drought and its effects on plants. The authors should use additions investigations and citations in the introduction to make it deeply written. Please cite this recent and related published papers about drought stress:
https://doi.org/10.3390/plants10020259
https://doi.org/10.1016/j.jenvman.2021.113076
https://doi.org/10.1038/s41598-020-73489-z
I do not recommend authors to use Figures in introduction, so please remove Figure 1 from the introduction. Anyone can search about it on the internet.
Material and Methods
L119-120 why the authors wrote the statistical analysis part here? These should be merged with the text in section 2.4. statistical analysis. Please move this there…
L111-120 The methods should be well described in details instead of writing few words for each parameter in section 2.2 How it was done? What chemicals were used? What instruments were used with more details for company, country of made etc?
Results:
It was well written according to the treatments of the study, but I recommend the authors to move some figures into supplementary materials.
Discussion:
Although the authors made it good but they should cite the text where it is possible. In addition, they always should link the interpretation of the cited publication with their results.
L447-447 please cite this ref. https://doi.org/10.1016/j.agwat.2020.106626
What is the difference between drought stress and water stress or water deficit? The authors used these different terms many times within the ms, they should be careful!
Conclusion: Please make it shorter and write only the most important findings. You do not have to repeat yourself again and again.
Regards, Reviewer
Author Response
reviewer 2:
Comments and Suggestions for Authors
Comment 1: The ms sustainability-1713088 with the title of Physiological and Transcriptomic Responses of Illicium difengpi to Drought Stress investigated the physiological index determination and transcriptome sequencing experiments in biennial Illicium difengpi seedlings grown under different soil moisture conditions (70~80%, 40~50% and 10~20%) to elucidate the molecular mechanisms of the response to drought stress in Illicium difengpi. The ms have to be significantly improved before it can go for father process.
Response 1: Thank you very much for your valuable comments and suggestions which are very helpful to improve our manuscript. Following these suggestions, we have supplemented the “Introduction” and “Materials and methods” sections of the manuscript and made significant revisions to the “Conclusions” section. In addition, appropriate revisions have also been made in the “Results” and “Discussion” sections. All the revised portions are marked in yellow in the manuscript.
Abstract:
Comment 2:The authors should make it shorter by reducing the text in the first two sentences (background) and mention only the key results.
Response 2: Thanks again for your comment, we have make the abstract shorter by reducing the text in the first two sentences (background); simultaneously, We have also streamlined other parts of the abstract and mention only the key results.
Comment 3: The variations ((70~80%, 40~50% and 10~20%) or the range for the soil moisture conditions within each treatment is big (10%) and this can cause a mistake in the obtained results. 10% range is high error, and the authors should have made it accurate to get accurate results.
Response 3: During the experiment, the soil moisture content determination in different treatments were conducted, and the results showed that the soil moisture contents of different treatments were 74.27±2.03% (70~80%), 47.37±3.04% (40~50%) and 17.54±1.76% (10~20%) respectively. The variations for the soil moisture conditions within each treatment are smaller than that of reported by the previous study (https://doi.org/10.1038/s41598-020-73489-z).
Comment 4: The treatments are very simple and I was looking to see that the authors used some treatments to mitigate the water or drought stress. But, this was not found.
Response 4: In this study, our purpose was to preliminarily explore the molecular mechanism of Illicium difengpi response to different drought conditions, so we did not carry out the experiment related to mitigate the water or drought stress. Nonetheless, the rehydration experiments of I. difengpi under drought stress are under way, the results of which will be reported in the next paper.
Introduction:
Comment 5: The authors wrote long text about the I. difengpi, but they did not write deeply about the drought stress. They should write one more paragraph of few sentences about the drought and its effects on plants. The authors should use additions investigations and citations in the introduction to make it deeply written. Please cite this recent and related published papers about drought stress:
https://doi.org/10.3390/plants10020259
https://doi.org/10.1016/j.jenvman.2021.113076
https://doi.org/10.1038/s41598-020-73489-z
Response 5: Thank you for recommending references for us, these references are helpful to make our manuscript deeply written. Citing these references, we have written about the drought stress in the first paragraph of the introduction.
Comment 6: I do not recommend authors to use Figures in introduction, so please remove Figure 1 from the introduction. Anyone can search about it on the internet.
Response 6: According to your suggestion, we have removed Figure 1 from the introduction.
Material and Methods
Comment 7: L119-120 why the authors wrote the statistical analysis part here? These should be merged with the text in section 2.4. statistical analysis. Please move this there…
Response 7: We apologize for the occurrence of such low-level errors. According to your suggestion, we have merged the statistical analysis on line 119-120 with the text in section 2.4.
Comment 8: L111-120 The methods should be well described in details instead of writing few words for each parameter in section 2.2 How it was done? What chemicals were used? What instruments were used with more details for company, country of made etc?
Response 8: Thank you for your reminder. In the revised manuscript, we have described in detail the experimental methods for the determination of physiological indicators. All supplements have been marked in yellow.
Results:
Comment 9: It was well written according to the treatments of the study, but I recommend the authors to move some figures into supplementary materials.
Response 9: We have moved the figure 4 and figure 5 into supplementary materials, and the new numbers of which are figure s1 and figure s2, respectively.
Discussion:
Comment 10: Although the authors made it good but they should cite the text where it is possible. In addition, they always should link the interpretation of the cited publication with their results.
L447-447 please cite this ref. https://doi.org/10.1016/j.agwat.2020.106626
Response 10: Thank you for recommending references for us, we have cited the ref. https://doi.org/10.1016/j.agwat.2020.106626 in first paragraph of the “4. Discussion” section, and you can see it in the yellow marked section.
Comment 11: What is the difference between drought stress and water stress or water deficit? The authors used these different terms many times within the ms, they should be careful!
Response 11: The meaning of drought stress and water stress or water deficit in this manuscript is the same. In order to unify the expression, we have revised the water stress and water deficit into drought stress.
Comment 12: Conclusion: Please make it shorter and write only the most important findings. You do not have to repeat yourself again and again.
Response 12: We have rewritten the conclusions section to succinctly describe the results, significance and future perspectives or hypothesize of this study.
Reviewer 3 Report
The article entitled " Physiological and Transcriptomic Responses of Illicium difengpi to Drought Stress". The article was aimed to elucidate the molecular mechanisms of the response to drought stress in Illicium difengpi, physiological index determination and transcriptome sequencing experiments were conducted in biennial I. difengpi seedlings grown under different soil moisture conditions. Study results achieved the aim of the study through physiological analyses proline (Pro), soluble sugar (SS) and malondialdehyde (MDA) contents increased; superoxide dismutase (SOD) and peroxidase (POD) activities and transcriptome sequencing followed by bioinformatic analyses.
The manuscript comprises all the necessary elements of scientific paper. I recommend this article for publication after incorporating minor changes given in below.
Introduction looks shallow. It can be improved for better understanding.
Organisms and their scientific name should be full in the first mention all others are should be abbreviated. Authors have mentioned the full form of scientific name throughout the manuscript. Illicium difengpi --> I. difengpi.
The methods section should contain a framework figure of listing all the analysis/steps done in this paper.
Authors should state whether experiments were conducted in technical triplicates or biological duplicates in statistical analysis section.
Please change the figures 2 and 3 as color.
In conclusion, write few lines about the future perspectives or hypothesize about the study. Discuss more and it will be useful to the readers for ease of understanding.
In materials and method section: authors should explain why each item of methodology was done.
qRT-PCR related information is lacking in Materials and methods section authors should focus on the same and rectify it.
Please describe more about in the RNA extraction section in the materials and method section.
Authors must concentrate on the formatting, and use of symbols, etc., in throughout manuscript.
Author Response
reviewer 3:
Comments and Suggestions for Authors
Comment 1: The article entitled " Physiological and Transcriptomic Responses of Illicium difengpi to Drought Stress". The article was aimed to elucidate the molecular mechanisms of the response to drought stress in Illicium difengpi, physiological index determination and transcriptome sequencing experiments were conducted in biennial I. difengpi seedlings grown under different soil moisture conditions. Study results achieved the aim of the study through physiological analyses proline (Pro), soluble sugar (SS) and malondialdehyde (MDA) contents increased; superoxide dismutase (SOD) and peroxidase (POD) activities and transcriptome sequencing followed by bioinformatic analyses.
The manuscript comprises all the necessary elements of scientific paper. I recommend this article for publication after incorporating minor changes given in below.
Response 1: Thank you very much for your valuable comments and suggestions which are very helpful to improve our manuscript. Based on your comments, we have made a revision in the manuscript, which you can see in the yellow marked sections.
Comment 2: Introduction looks shallow. It can be improved for better understanding.
Response 2: In introduction, we have added something about the drought stress and the effects of which on the plants; simultaneously, we also made appropriate modifications to the description of the I. difengpi habitat. All these revision make the introduction section more closely related to our research.
Comment 3: Organisms and their scientific name should be full in the first mention all others are should be abbreviated. Authors have mentioned the full form of scientific name throughout the manuscript. Illicium difengpi --> I. difengpi.
Response 3: Thank you for your comments, we have revised the Illicium difengpi into I. difengpi throughout the manuscript except for the first mention.
Comment 4: The methods section should contain a framework figure of listing all the analysis/steps done in this paper.
Response 4: We have added a framework figure in the revised manuscript, and you can see it in the figure 1.
Comment 5: Authors should state whether experiments were conducted in technical triplicates or biological duplicates in statistical analysis section.
Response 5: Thank you for your reminder. In our research, the determination of physiological indicators and transcriptome sequencing were conducted in three biological duplicates, the statements of which you can see on the third line of “2.2. Determination of Physiological Indicators” and the first line of “2.3.1. RNA Extraction, Library Construction and Sequencing”. The reason why we put the statements in the experimental method rather than in the statistical analysis section is to make the experimental design more complete and avoid the repetition in the text.
Comment 6: Please change the figures 2 and 3 as color.
Response 6: We have changed the figures 2 and 3 as color.
Comment 7: In conclusion, write few lines about the future perspectives or hypothesize about the study. Discuss more and it will be useful to the readers for ease of understanding.
Response 7: Thank you for your suggestion, we written few lines about the future perspectives or hypothesize about the study, which you can see in the conclusion section (In the future, we will conduct multi-omics analysis……the species in ecological restoration in karst areas).
Comment 8: In materials and method section: authors should explain why each item of methodology was done.
Response 8: In materials and method section: we have explained the reasons for using these methodologies in places that are not easily understood by the reader. But in some places, where can be apprehended at a glance by readers we have not give the explanations. The specific explanation is as follows:
- On line 1 of the “2.3.2 Raw Data Processing”, we added “In order to facilitate the analysis, publication and sharing of sequencing data…”.
- On line 3-4 of the “2.3.2 Raw Data Processing”, we added “To ensure the accuracy of the subsequent bioinformatics analysis,”.
- On line 1 of the “2.3.3 Annotation and Gene Expression Analysis”, we added “To explore the biological functions of the assembled transcripts and unigenes,”.
- On line 5-7 of the “2.3.4 Screening and Enrichment Analysis of DEGs”, we added “In order to study the differences in molecular mechanisms of I. difengpi response to different drought stresses,”.
5.On line 1 of the “2.3.5 Quantitative Real-time PCR (qRT–PCR) Analysis”, we added “In order to verify the accuracy of the RNA-seq data”.
Comment 9: qRT-PCR related information is lacking in Materials and methods section authors should focus on the same and rectify it.
Response 9: Thank you very much for your comment, we have added the “2.3.5 Quantitative Real-time PCR (qRT–PCR) Analysis” in the revised manuscript to descript the experiment methods of qRT–PCR.
Comment 10: Please describe more about in the RNA extraction section in the materials and method section.
Response 10: Since the experimental steps of RNA extraction are complex, it would be cumbersome to describe in detail in the text, so we provide the information of the kits used and the basis for the experimental operation. The revision you can see in the “2.3.1. RNA Extraction, Library Construction and Sequencing” section (RNA extraction was performed with TRIzol® Reagent (Invitrogen), following the manufacturer's instructions).
Comment 11: Authors must concentrate on the formatting, and use of symbols, etc., in throughout manuscript.
Response 11: I apologize for the occurrence of such low-level errors. We have adjusted the full text format, and use of symbols, etc., in throughout manuscript.
Round 2
Reviewer 1 Report
The manuscript is substantially improved.
There are, however, a few minor points that should be addressed.
P-adjusted values. You have now clarified that you have used four methods (Bonferroni, Holm, Sidak and false discovery rate). But I see ONE column of P-adjust, So, it is not clear which method was used for the reported P-adjust values
Figures 2 and 3. Please, explain what the error bars refer to. Standard errors? Standard deviations?
Lines 288, 298, 303. I assume that these p-values refer to Duncan's post hoc tests following ANOVA. I suggest reporting this to help the reader. It may be sufficient expressions like: “difengpi (Duncan's post hoc tests following ANOVA: p < 0.05) (Figure 3 c-e)”….” among different light levels (Duncan's post hoc tests following ANOVA: P < 0.05).”
Line 302: above columns -> above bars
377-384. Revise grammar. You say “We found that…” but there is no verb
Figure 8. These are percent numbers. Please, change the caption to “Percent number of ….”
529: . FC, the fold -> . FC is the fold
Probability is indicates sometimes with P sometimes with p. Please, check for consistence.
Author Response
reviewer 1:
Comments and Suggestions for Authors
Comment 1: The manuscript is substantially improved. There are, however, a few minor points that should be addressed.
Response 1: Thanks again for your comments and suggestions, which are very helpful to improve our manuscript. We will take every comment and suggestion you give seriously. All the revised portions are marked in yellow in the revised manuscript.
Comment 2: P-adjusted values. You have now clarified that you have used four methods (Bonferroni, Holm, Sidak and false discovery rate). But I see ONE column of P-adjust, So, it is not clear which method was used for the reported P-adjust values.
Response 2: I apologize for the last reply failed to explain the problem clearly. After communicating with the Major Biomedical Technology Co., Ltd. (China), a company that provides us with transcriptome sequencing services, we clarified that the p-value was adjusted by using the method of FDR (false discovery rate) in this study.
Comment 3: Figures 2 and 3. Please, explain what the error bars refer to. Standard errors? Standard deviations?
Response 3: The error bars in Figures 2 and 3 refer to standard deviations.
Comment 4: Lines 288, 298, 303. I assume that these p-values refer to Duncan's post hoc tests following ANOVA. I suggest reporting this to help the reader. It may be sufficient expressions like: “difengpi (Duncan's post hoc tests following ANOVA: p < 0.05) (Figure 3 c-e)”….” among different light levels (Duncan's post hoc tests following ANOVA: P < 0.05).”
Response 4: We appreciate your suggestions and fully adopt them in the revised version. All the revisions you can see in the revised manuscript.
Comment 5: Line 302: above columns -> above bars
Response 5: I apologize for the occurrence of such low-level errors. We have made a modification in the revised manuscript.
Comment 6: 377-384. Revise grammar. You say “We found that…” but there is no verb
Response 6: We have modified this sentence according to your comment, and you can see it in the fifth paragraph of “3.4. Classification of DEGs” section. ( Here, we found that the early upregulated genes showed significant enrichment in 4 pathways,…and “drug metabolism - cytochrome P450” (Figure S4 B))
Comment 7: Figure 8. These are percent numbers. Please, change the caption to “Percent number of ….”
Response 7: In figure 8, I think you may have misunderstood the numbers provided in the figure. Actually, the numbers to the right of the bars refer to the number of DEGs encoding transcription factors proteins, not the percent numbers. In total, 244 TFs from 10 families showing differential expression under light stress or severe stress were identified.
Comment 8: 529: . FC, the fold -> . FC is the fold
Response 8: We have made a modification in the revised manuscript according to your suggestion.
Comment 9: Probability is indicates sometimes with P sometimes with p. Please, check for consistence.
Response 9: Thanks for your reminder. We have checked the consistency of the format of “P” in the text and made modifications.

Reviewer 2 Report
The ms wa improved and it can be accepted now.
Thanks for authors
Good luck
Author Response
Thank you for your comments and suggetions again,which are very helpful for revising and improving our manuscript.